# Assessment of the Photosynthetic Apparatus Functions by Chlorophyll Fluorescence and P_700_ Absorbance in C3 and C4 Plants under Physiological Conditions and under Salt Stress

**DOI:** 10.3390/ijms23073768

**Published:** 2022-03-29

**Authors:** Martin A. Stefanov, Georgi D. Rashkov, Emilia L. Apostolova

**Affiliations:** Institute of Biophysics and Biomedical Engineering, Bulgarian Academy of Sciences, Acad. G. Bonchev Str., Bl. 21, 1113 Sofia, Bulgaria; martin_12.1989@abv.bg (M.A.S.); megajorko@abv.bg (G.D.R.)

**Keywords:** NaCl treatment, JIP test, PAM chlorophyll fluorescence, pea, photooxidation of P_700_, maize

## Abstract

Functions of the photosynthetic apparatus of C3 (*Pisum sativum* L.) and C4 (*Zea mays* L.) plants under physiological conditions and after treatment with different NaCl concentrations (0–200 mM) were investigated using chlorophyll a fluorescence (pulse-amplitude-modulated (PAM) and JIP test) and P_700_ photooxidation measurement. Data revealed lower density of the photosynthetic structures (RC/CSo), larger relative size of the plastoquinone (PQ) pool (N) and higher electron transport capacity and photosynthetic rate (parameter R_Fd_) in C4 than in C3 plants. Furthermore, the differences were observed between the two studied species in the parameters characterizing the possibility of reduction in the photosystem (PSI) end acceptors (REo/RC, REo/CSo and δRo). Data revealed that NaCl treatment caused a decrease in the density of the photosynthetic structures and relative size of the PQ pool as well as decrease in the electron transport to the PSI end electron acceptors and the probability of their reduction as well as an increase in the thermal dissipation. The effects were stronger in pea than in maize. The enhanced energy losses after high salt treatment in maize were mainly from the increase in the regulated energy losses (Φ_NPQ_), while in pea from the increase in non-regulated energy losses (Φ_NO_). The reduction in the electron transport from Q_A_ to the PSI end electron acceptors influenced PSI activity. Analysis of the P_700_ photooxidation and its decay kinetics revealed an influence of two PSI populations in pea after treatment with 150 mM and 200 mM NaCl, while in maize the negligible changes were registered only at 200 mM NaCl. The experimental results clearly show less salt tolerance of pea than maize.

## 1. Introduction

Plants in their development are exposed to various environmental stresses. Salinity is one of the most widespread environmental factors, which limits plant growth and crop productivity [1,2]. Toxic sodium levels in plants harm biological membranes and subcellular organelles, causing an inhibition of biochemical and physiological processes [3]. The harmful effects of salinity on the plant are caused by osmotic stress and ionic stress, and subsequent oxidative damage often occurs [4,5]. Oxidative damage of lipids and proteins leads to damage of the photosynthetic apparatus and a decrease in the photosynthetic activity [6,7,8]. The enhanced salt-induced reactive oxygen species (ROS) cause lipid peroxidation leading to increased membrane fluidity and permeability [9]. High saline conditions also lead to a reduction in the total pigment content and impact on the chloroplast structure [10,11,12]. It has been shown that salinization leads to swelling of chloroplasts, increasing the number and size of plastoglobuli, destroying of thylakoid membranes in these organelles, and subsequent destruction of the structure of these photosynthetic organelles [8,13,14,15,16]. These changes inhibit the functions of the photosynthetic electron transport chain, which depend on the plant species [12,17]. Salinity affects photosystem II (PSII) more than photosystem I (PSI) [10]. The inhibition of PSII functions is a result of disintegrating the PSII reaction center, the oxygen-evolving complex (OEC) and by limiting the activity of quinone acceptors [10]. The influence of the salt stress on the photosynthetic components, as well as the extent of damage, is determined by the strength and duration of the stress [12,18].

The most common plants can be divided into two major groups, as C3 photosynthesis and C4 photosynthesis, depending on the first product of the CO_2_ fixation [19,20,21]. The photosynthetic CO_2_ assimilation rates, the photosynthetic efficiency, the biomass production and the efficient use of light in C4 plants are higher than in C3 plants [22,23]. Previous studies also revealed that the ratios chlorophyll *a/b*, lipid/chlorophyll and lipid/protein, as well as the mobility of pigment–protein complexes in thylakoid membranes of C3 and C4 plants are different [24,25,26,27]. Furthermore, the plants with these two photosynthetic pathways respond quite differently to changes in the environmental conditions. Differences in antioxidant defense under drought and salinity in C3 and C4 plants have also been shown [28,29].

Chlorophyll fluorescence is widely used to study the influence of the abiotic stress factors on photosynthetic performance [30] and gives information about the energy transfer in the photosynthetic apparatus and related photosynthetic processes [31,32,33,34,35,36]. The widespread use of chlorophyll fluorescence is determined by the fact that it is a fast, informative and non-invasive method. One of the most widely used measurement techniques for characterization of the functions of photosynthetic apparatus is pulse-amplitude modulated (PAM) and chlorophyll *a* fluorescence induction. The parameters of PAM chlorophyll fluorescence can be used to determine both the structural differences between dark-adapted state and light-adapted state and the photosynthetic performance at light-adapted state [37]. A rapid increase in the chlorophyll fluorescence in dark-adapted plants after exposure to light (JIP parameters) describes the primary photosynthetic reactions and evaluates important PSII characteristics, such as energy trapping, electron transport and dissipation of the excitation energy in the antenna complexes [38,39]. In recent years, the JIP test has been used for understanding the mechanism of action of different stress factors on the plants.

We hypothesized that differences in the structure, functions and antioxidant activity of the C3 and C4 plants would differentially affect the activity of different components of the photosynthetic apparatus under salt stress. Using chlorophyll fluorescence (PAM and chlorophyll *a* fluorescence induction), we evaluated the effects of different concentrations of NaCl on the primary processes of photosynthesis in pea (*Pisum sativum* L.) and in maize (*Zea mays* L.). Data clearly show the influence of NaCl concentration on the various components of the electron transport chain, as well as the higher sensitivity of pea compared with maize at different levels of salinization.

## 2. Results

### 2.1. Effects of Salinity on the Chlorophyll a Fluorescence

Analysis of the PAM chlorophyll fluorescence signals revealed a strong influence on the maximum quantum yields of primary photochemistry of PSII (Fv/Fm), the ratio of photochemical to non-photochemical processes in PSII (Fv/Fo), the photochemical quenching (q_p_) and the PSII based electron transport rate (ETR(II)) in both studied species exposed to the highest NaCl concentration (200 mM NaCl) (Figure 1 and Figure 2). The effects on these parameters were stronger in pea than in maize. The decrease in the ETR(II) was 90% in pea and 65% in maize, while q_p_ decreased by 72% and 30% in pea and maize, respectively (Figure 1 and Figure 2). Experimental results also reveal that after treatment with 150 mM, NaCl changes in Fv/Fm, Fv/Fo, q_p_ and ETR(II) were registered only in pea. In addition, data reveal that the changes in the parameters Fv/Fm and Fv/Fo were smaller than ETR(II) and q_p_. Data also demonstrate that the values of the studied PAM parameters (Fv/Fm, Fv/Fo, q_p_ and ETR(II)) after the treatment with 50 mM NaCl were similar to the untreated plants in both studied species.

The impact of different levels of salinity on the light energy utilization in PSII in the C3 and C4 plants was determined by the effective quantum yield of the photochemical energy conversion in PSII (Φ_PSII_) and the quantum yields of regulated (Φ_NPQ_) and non-regulated (Φ_NO_) energy losses in PSII. The sum of the parameters Φ_PSII_, Φ_NPQ_ and Φ_NO_ is equal to 1 [49,50]. High concentrations of NaCl (150–200 mM) more strongly affected the parameter Φ_PSII_ in pea than in maize (Figure 3). After NaCl treatment this parameter in pea decreased by 25% at 150 mM and 90% at 200 mM, while in maize the decrease was from 18% (at 150 mM) to 65% (at 200 mM). Data also reveal that the influence on the quantum yields of regulated (Φ_NPQ_) and non-regulated (Φ_NO_) energy losses in PSII after NaCl treatment was different in the studied plants. The enhancement of the energy losses in pea was connected mainly with an increase in the Φ_NO_, while in maize with an increase in Φ_NPQ_ (Figure 3).

The non-photochemical quenching of chlorophyll fluorescence (NPQ) is an important photoprotective mechanism in plants [51]. It has been shown that NPQ involves three components: the energy-dependent quenching (qE), mediated by the proton gradient across the thylakoid membrane; the state transition quenching (qT), induced by the reversible phosphorylation of the light-harvesting complex of PSII (LHCII); and the photoinhibitory quenching (qI)) [52]. The impact of different levels of salinity on the quantum yields of these components (ΦqE, ΦqT and ΦqI) is shown in Figure 4. The values of components ΦqE and ΦqT were higher in maize compared with pea, but values of ΦqI were higher in pea than in maize in the treated and untreated plants. In addition, data revealed an increase in the ΦqI in pea treated with all NaCl concentrations, while in maize an increase in this component was registered only at 200 mM NaCl. The components ΦqE and ΦqT increased after treatment with higher NaCl concentrations (150 mM and 200 mM) in both studied plant species, but the changes were more pronounced in pea than in maize (Figure 4).

The kinetic of dark relaxation of chlorophyll fluorescence induced by a single saturating light pulse in dark-adapted leaves gives information for the electron transfer between Q_A_ and plastoquinone [53,54]. The fluorescence signals could be fitted to two components with fast (k_1_) and slow (k_2_) rate constant in treated and untreated plants of pea and maize. These constants were less influenced in maize than in pea after treatment with NaCl (Figure 5). In maize, k_1_ was influenced only after treatment with 200 mM NaCl, while k_2_ was similar to control plants. Data also reveal that in pea both constants decreased after treatment with 150 mM and 200 mM NaCl, as the effect was more pronounced for constant k_2_ (Figure 5).

### 2.2. Effects of Salinity on the Rate of Photosynthesis

The chlorophyll fluorescence decay ratio (R_Fd_), which correlates with the net assimilation of CO_2_ [55], was used to assess the influence of salinity on the rate of photosynthesis.

Salt-induced changes in the photosynthetic apparatus led to a decrease in the R_Fd_ parameter in both studied species (Figure 6). Data reveal that the parameter R_Fd_ decreases in pea treated with 150 mM NaCl (by 29%) and 200 mM NaCl (by 84%), while in maize an influence was observed only at the highest NaCl concentration (at 200 mM NaCl, decreases by 20%) (Figure 6).

### 2.3. Effects of Salinity on the Chlorophyll Fluorescence Induction

For more information about the influence of the different levels of salinity on the photosynthetic apparatus, selected JIP parameters were calculated (Table 1).

Comparison of two studied plants under physiological conditions revealed differences in parameters ETo/RC, REo/RC, REo/CSo, TRo/CSo, RC/CSo φEo, φRo, ψEo, δRo and N (Figure 7). The data show a low density of the photosynthetic structure (RC/CSo), but a larger relative size of the PQ pool per reaction center (N) and more efficient operation of PSI in the maize compared with the pea. The value for REo/RC was 3 times higher in maize than pea. In addition, the data reveal that parameters which characterized quantum efficiency of the processes in the photosynthetic apparatus (φEo, φRo, ψEo) also had higher values in maize than in pea—i.e., the electron transport efficiency is higher in C4 than in C3 plants. The differences between the studied plants were larger in electron transport flux until the PSI end electron acceptors (REo/RC) and quantum yields for reduction in PSI end electron acceptors (φRo).

Significant differences between the studied untreated plants were found in the performance indices (PI_ABS_, PI total), as the values of these parameters were much higher in maize compared with pea (Figure 7). The index PI_ABS_ includes three parameters: the number of active RC per PSII antenna chlorophyll [γRC2/(1 − γRC2) = RC/ABS], the partial performance of primary photochemistry [φPo/(1 − φPo)] and the performance of thermal reactions of the intersystem electron carriers [ψEo/(1 − ψEo)] [35,41,56]. Higher values of PI_ABS_ in maize were the result of the high value of the third component [ψEo/(1 − ψEo)], which characterizes non-light-dependent reactions (Table 2). The observed differences in the PI total between pea and maize were the result of the differences in the efficiency/probability with which an electron is transferred from Q_B_ to PSI electron acceptors [(δREo/(1 − δREo)] (Table 2). The values of this component were about 3 times higher in the maize than the pea.

Equations for performance indices:PIABS = RC/ABS × φPo/(1 − φPo) × ψEo/(1 − ψEo);
PI total = PI_ABS_ × δREo/(1 − δREo)

After treatment with 50 mM NaCl, the values of the measured JIP parameters (Table 1) in maize were similar to the values of the control plants, while negligible effects on them in pea were observed. Data revealed a small decrease in the parameters REo/RC, REo/CSo, φRo, δRo and N in the pea than in the control, while in the maize the values of these parameters are similar to the untreated plants (Figure 8).

The treatment with 150 mM and 200 mM NaCl influenced all studied parameters, as the effects were more pronounced in the pea compared with the maize (Figure 8 and Figure 9). After treatment of pea with 150 mM NaCl, changes were observed in the electron transport capacity (φEo, φRo, δRo, REo/CSo), absorbance flux per cross-section (ABS/CSo) and the processes related to energy dissipation (DIo/RC and DIo/CSo) (Figure 8 and Figure 9). At the same time, the changes in these parameters were negligible in the maize. In addition, the changes in parameters Vj and Wk were registered in both studied plants after treatment with 200 mM NaCl (Figure 2). The increased value of Wk indicates damage of the OEC after salt treatment. An increase in the values of Vj after treatment with 150 mM NaCl was observed only in pea (Figure 2). After treatment with the highest NaCl concentration (200 mM), the effects on the JIP parameters were stronger in the pea than in the maize (Figure 2, Figure 8 and Figure 9).

Treatment with high salt concentration led to a stronger influence of salinity on the performance index (PI total) measuring the energy conservation from exciton to the reduction in PSI end acceptor) than the performance index on the absorption base (PI_ABS_, energy conservation from exciton to the reduction in intersystem electron acceptors) (Figure 8). The data reveal that PI total after treatment with 200 mM NaCl decreased by 98% and 73% for pea and maize, respectively. Performance index PI total (45%) also decreased (45%) in pea after treatment with 150 mM NaCl. The performance index PI_ABS_ was also less affected by salinity in maize (decreases with 11% at 150 mM and 63% at 200 mM NaCl) than in pea (21% and 99% decrease for 150 mM and 200 mM NaCl, respectively) (Figure 8).

### 2.4. P_700_ Photooxidation

Additional information for the influence of salinity on PSI function was provided by the study of P_700_ photooxidation by far-red light. The far-red light-induced steady-state oxidation of P_700_ (ΔA/A) and the times *t*_1_ and *t*_2_ of the P_700_^+^ dark reduction were calculated (Figure 10). The parameter ΔA/A in pea decreased after treatment with high NaCl concentrations (150 mM and 200 mM), while in maize the values at these concentrations were similar to the control. The dark-reduction in P_700_^+^ was deconvoluted into two exponential components with times *t*_1_ and *t*_2_ for the fast and the slow exponents, respectively. The influence of higher NaCl concentrations on the *t*_1_ and *t*_2_ was less in maize than pea. In addition, data revealed a decrease in *t*_1_ in maize after treatment with 50 mM (by 21%) and 150 mM NaCl (by 27%), while in pea the decrease in *t*_1_ (by 33%) was registered only after treatment with 50 mM NaCl (Figure 10). At the same time, a decrease in the *t*_2_ was registered after treatment with 200 mM in maize plants, while in pea an increase in *t*_2_ was observed after treatment with 150 mM and 200 mM NaCl (Figure 10).

## 3. Discussion

The influence of salinity on plants is a complex phenomenon. This study for the first time focused on the effects of different NaCl concentrations on the photosynthetic apparatus in C3 and C4 plants. *Pisum sativum* L. was used as a model for C3 plants, and *Zea mays* L. was chosen to represent C4 plants.

Previous studies and the present study revealed some differences in the amount of chlorophyll and Chl *a/b* ratio in pea and maize under physiological conditions [24,25,26,27,57] and Appendix A. Based on these studies, different organizations of the thylakoid membranes in C3 and C4 plants could be suggested. It was also shown that the smaller Chl *a/b* ratio corresponds with the increased amount of chlorophyll in grana and suggests the synthesis of the additional LHCII molecules [58]. Furthermore, the differences in lipid/chlorophyll and in lipid/protein rations in C3 and C4 plants influence the mobility of pigment–protein complexes in the thylakoid membranes [25]. Data in the current study reveal that the value of the widely used parameter Fv/Fm that assesses PSII activity was similar in both studied plant species (Figure 1). The value of this parameter in higher plants is around 0.8 [59,60]. Previous studies have shown a lower or similar value of Fv/Fm ratio in C4 compared with C3 plants [61]. These differences could be due to plant growing conditions as well as to plant species. On the other hand, the reduction in the chlorophyll content could not be the cause for a decrease in the parameter Fv/Fm, which indicates that pigment amount is not associated with the maximum quantum yield of the primary PSII photochemistry. Similar results have been shown in the study of Guha et al. [39]. At the same time, our data reveal a higher photosynthetic rate (parameter R_Fd_) in maize than in pea (Figure 6). This observation is in agreement with the previous studies [22,23,62]. In addition to these observations, the experimental results in this study reveal variation in the functional efficiency of different components of the photosynthetic apparatus in studied plants, which could be a result of differences in the density of the photosynthetic structure (RC/CSo the relative size of the PQ pool (N) and electron transport activity in maize and pea (Figure 7).

Previous investigations revealed the effects of the salinity on the chlorophyll content, the structure of the chloroplast, membrane injury and enhanced amount of ROS [63,64,65]. It has been also shown that high salt concentrations led to a restriction of the electron flow from Q_A_^−^ to the plastoquinone pool, influencing Q_A_^−^ reoxidation by plastoquinone and by the recombination of electrons in Q_A_Q_B_^−^ via the Q_A_^−^Q_B_ ↔ Q_A_Q_B_^−^ charge equilibrium with the oxidized S_2_ (or S_3_) state of the OEC [24,45,66]. The results of the present study also show an influence of the salinity on the constants (k_1_ and k_2_) characterizing the Fm decay (Figure 5)—i.e., influence on the interaction between Q_A_ and PQ. The impact of salinity on these constants was stronger in pea than in maize. All of these salt-induced changes cause a decrease in the efficiency of the photosynthetic machinery. Previous investigations [24,45,67] and the experimental results of the present study revealed a decrease in the potential of PSII efficiency (Fv/Fm) as well as in the ratio of the photochemical to nonphotochemical processes (Fv/Fo) (Figure 1 and Figure 2). Having in mind that the ratio Fv/Fo corresponds to the efficiency of the OEC [68,69,70], it can be concluded that the high salt concentrations induced changes in the donor side of PSII, which were larger in pea than in maize plants. This statement is supported by the changes in the parameter Wk, indicating activity of the OEC (Figure 2). A strong increase in the Wk was found in both studied species after treatment with the highest NaCl concentration (Figure 2), which indicates dissociation of this complex and its damage [71]. Some authors suggest a correlation between changes in this parameter and salt sensitivity of the plants [43]. The suggested modification of the OEC was confirmed by previous study, which revealed that salt-induced changes in the donor side of PSII [72,73]. Data in this study also revealed an increase in the Vj and decrease in the ψEo in both studied plants after treatment with 150 mM and 200 mM NaCl (Figure 2 and Figure 8), which could be a result of the accumulation of reduced Q_A_ and limitation of the electron transport beyond Q_A_ [74]. In support of this assumption for changes in the PSII acceptor side are the changes in the constants k_1_ and k_2_ characterizing the Fm decay (Figure 5), as well as previous studies [24,45,66,72,73].

The changes in the photosynthetic apparatus after salt treatment led to a decrease in the open reaction centers of PSII (qp) and an inhibition of the ETR(II), as these parameters changed differently in pea and maize (Figure 1). The decrease in the efficiency of the open reaction centers [24] and an increase in the closed reaction centers after NaCl treatment (qp decrease, Figure 1) resulted in a decrease in the Φ_PSII_ and an increase in the quantum yields of regulated (Φ_NPQ_) and non-regulated (Φ_NO_) energy losses in PSII (Figure 3). The increase in non-regulated energy losses due to PSII inactivation and thermal dissipation indicates an increase in the ROS production, which causes damage of PSII [49,75,76]. The changes in energy losses in maize were mainly due to an increase in regulated energy losses (Φ_NPQ_), with the largest changes at 200 mM NaCl (Figure 3), but the values of Φ_NO_ were similar to the control. Salt-induced changes in pea after NaCl treatment led to a larger increase in Φ_NO_ in comparison with Φ_NPQ_. It is well known that non-photochemical quenching (NPQ) is the main photoprotective process that protects photosynthesis under abiotic stress [61]. More detailed information for the dissipation mechanism of the thylakoid membranes gives components of NPQ (qE, qT and qI) [48,77,78,79]. The energy-dependent (qE) and the state transition (qT) quenching are important for photoprotection of the photosynthetic apparatus, while the photoinhibitory quenching can be used to assess the PSII damage [78]. Our data reveal larger values of the qE and qT in maize plants than in pea plants, which could be one of the reasons for better tolerance to high salt concentrations in maize than in pea. Moreover, the strong increase in qI in pea after NaCl treatment suggests damage to the PSII complex (Figure 4).

The impact of salt-induced changes in thylakoid membranes on the PSI activity (the photooxidation of P_700_^+^ and its dark reduction kinetics) were different in studied species, and the changes varied in both studied plants depending on NaCl concentrations (Figure 10). It has been suggested that two components of P_700_^+^ decay (characterizing with time *t*_1_ and *t*_2_) originate from two electron-donor systems or due to two different populations of PSI located in different domains of the thylakoid membranes (in stroma lamellae and grana margin) [80,81]. The decrease in *t*_1_ corresponds to the stimulation of the cyclic electron transport around PSI, which plays a defensive role in preventing photosynthetic apparatus from oxidative damage under stress conditions [82]. Our data reveal that stimulation of the cyclic electron transport around PSI in pea was detected only at 50 mM NaCl, while in maize at 50 mM NaCl and 150 mM NaCl (Figure 10). The higher NaCl concentration led to a change in *t*_1_ and *t*_2_ in studied plant species, which suggests an influence on both populations of PSI. Furthermore, the data reveal stronger alteration in PSI in pea than in maize, which is associated with the stronger influence of the electron transport from Q_A_ to the PSI electron acceptors and the efficiency of their reduction (Figure 7).

## 4. Materials and Methods

### 4.1. Plant Growth Conditions and Treatments

The seedlings from pea (*Pisum Sativum* L. Ran1) and maize (*Zea mays* L. Method) were used in this study. Details of seedling growth are given in Stefanov et al. [24]. The plants were grown in a half-strength Hoagland solution. The cultivation of the plants was carried out in a photothermostat under controlled conditions: a 12 h light/dark photoperiod, a light intensity of 150 μmol photons/m^2^s, 28 °C (daily)/25 °C (night) temperature and 60% relative humidity. After 10 days of growth, NaCl (50 mM, 150 mM and 200 mM) was added to the nutrient solutions for 6 days. The plants that grew without NaCl were used as controls. Two independent experiments (20–25 uniform seedlings for each treatment, about 10 plants in pot) were performed for each treatment. For the analyses, we used the mature leaves (the middle part of the third and the fourth leaves).

### 4.2. Room-Temperature Chlorophyll Fluorescence

The pulse-amplitude-modulated (PAM) chlorophyll *a* fluorescence was measured using a PAM fluorometer (model 101/103, Walz GmbH, Effeltrich, Germany). Details for the measurements are described in Stefanov et al. [45]. The dark adaptation of leaves was 20 min. The maximum fluorescence levels in the dark-adapted (Fm) and light-adapted (Fm′) states were obtained with saturated pulses of 3000 μmol photons/m^2^s, which were provided by a Schott KL 1500 lamp (Schott Glaswerke, Mainz, Germany). The actinic light was 150 μmol photons/m^2^s. The following parameters were determined: the maximum quantum efficiency of PSII in dark-adapted state, Fv/Fm = (Fm − Fo)Fm; the photochemical quenching, qp = (Fm′ − Fs)/Fv′; the effective quantum yield of photochemical energy conversion of PSII, Φ_PSII_ = (Fm′ − Fs)/Fm′; the PSII based electron transport rate, ETR(II) = Φ_PSII_ × 150 × 0.5; the non-regulated (Φ_NO_ = Fs/Fm) and regulated (Φ_NPQ_ = Fs/Fm′ − Fs/Fm) energy loss in PSII; the ratio of quantum yields of photochemical and concurrent non-photochemical processes in PSII, Fv/Fo = (Fm − Fo)/Fo; [46,47]; the quantum yields of components of the non-photochemical quenching: the energy-dependent quenching, ΦqE, the state transition quenching, ΦqT, and the photoinhibitory quenching, ΦqI [48].

The constants of decay components (k_1_ for fast and k_2_ for slow) of the variable fluorescence relaxation after excitation by a saturated light pulse (3000 μmol photons/m^2^s) in dark-adapted leaves were determined [45,53,54].

The chlorophyll fluorescence decay ratio (R_Fd_ = Fd/Fs) was determined, where Fd is the fluorescence decrease from Fm to a steady state chlorophyll fluorescence (Fs) after continuous saturated illumination (3000 μmol photon/m^2^s). This ratio (R_Fd_) correlates with the net assimilation of CO_2_ [55].

Induction curves of chlorophyll fluorescence were measured with a Handy PEA+ instrument (Hansatech, Norfolk, UK) as described in Stefanov et al., [24]. The samples were dark-adapted for 20 min at room temperature using leaf clips. Prompt chlorophyll fluorescence was induced by strong light pulse (3000 μmol photons/m^2^s). All studied variants showed multiphase chlorophyll fluorescence increase during the first second of illumination after dark adaptation. The measured signals were used for calculations of the selected parameters of OJIP transitions (Table 1) [36,41,83,84].

All fluorescence measurements were performed on mature leaves (the middle part of the third and the fourth leaves).

### 4.3. P_700_ Photooxidation

The photooxidation of P_700_ (P_700_^+^) was measured on leaf discs, with a dual-wavelength (820 nm) unit (Walz ED 700DW-E) attached to a PAM101E main control unit in the reflectance mode. The details for the measurements are given in Dankov et al. [85]. The dark-adapted (20 min) leaf discs were illuminated with far-red light supplied by a photodiode (102-FR, Walz GmbH, Effeltrich, Germany). Changes in the oxidation of P_700_ (P_700_^+^) were assessed by red light-induced changes in absorbance at 820 nm (∆A). The ∆A/A ratio and the times of dark reduction in P_700_^+^ (*t*_1_ and *t*_2_) were calculated [45].

### 4.4. Statistical Analysis

Mean values ± SE were calculated from the data for at least two independent treatments with five biological replicates (five plants) of each variant. Statistically significant differences between the studied variants were identified using one-way ANOVA followed by a Tukey’s post hoc test for each parameter. Prior to the test, the assumptions for the normality of raw data (using the Shapiro–Wilk test) and the homogeneity of the variances (using Levene’s test) were checked. The homogeneity of variance test was used to verify the parametric distribution of data. Values were considered statistically different with *p* < 0.05 after Fisher’s least significant difference post hoc test by using Origin 9.0 software (OriginLab, Northampton, MA, USA).

## 5. Conclusions

In summary, the data reveal a larger relative size of the PQ pool and higher electron transport activity in maize compared with pea, as well as the lower density of the photosynthetic structure in maize than in pea. A significant difference in the electron transport capacity of the two studied species was observed in the parameters characterizing the possibility of reduction in the PSI end acceptors (REo/RC, REo/CSo and δRo). The treatment with higher NaCl concentrations led to: (i) a decrease in the density of the photosynthetic structure and relative size of the PQ pool; (ii) an increase in the apparent antenna size of an active PSII; (iii) a decrease in the efficiency and quantum yield for reduction in PSI end electron acceptors and increase in the thermal dissipation. All these changes were larger in C3 than C4 plants. Performance indices (PI_ABS_ and PI total), which are a sensitive indicator of the activity of the photosynthetic apparatus, were strongly reduced after treatment with high salt concentrations in both studied species, but the effects were more pronounced in C3 than in C4 plants. In addition, our data reveal enhanced energy losses after salt treatment in maize due to an increase in regulated energy losses (Φ_NPQ_), while in pea due to an increase in non-regulated energy losses (Φ_NO_). In conclusion, the data clearly show greater salt tolerance in maize than in pea. Furthermore, the experimental results in this study clearly show the high sensitivity of the processes in the photosynthetic apparatus in both C3 and C4 plants under different salinity levels; therefore, salt-induced changes in them could be used for determination of the salt tolerance of plants.

## Figures and Tables

**Figure 1 ijms-23-03768-f001:**
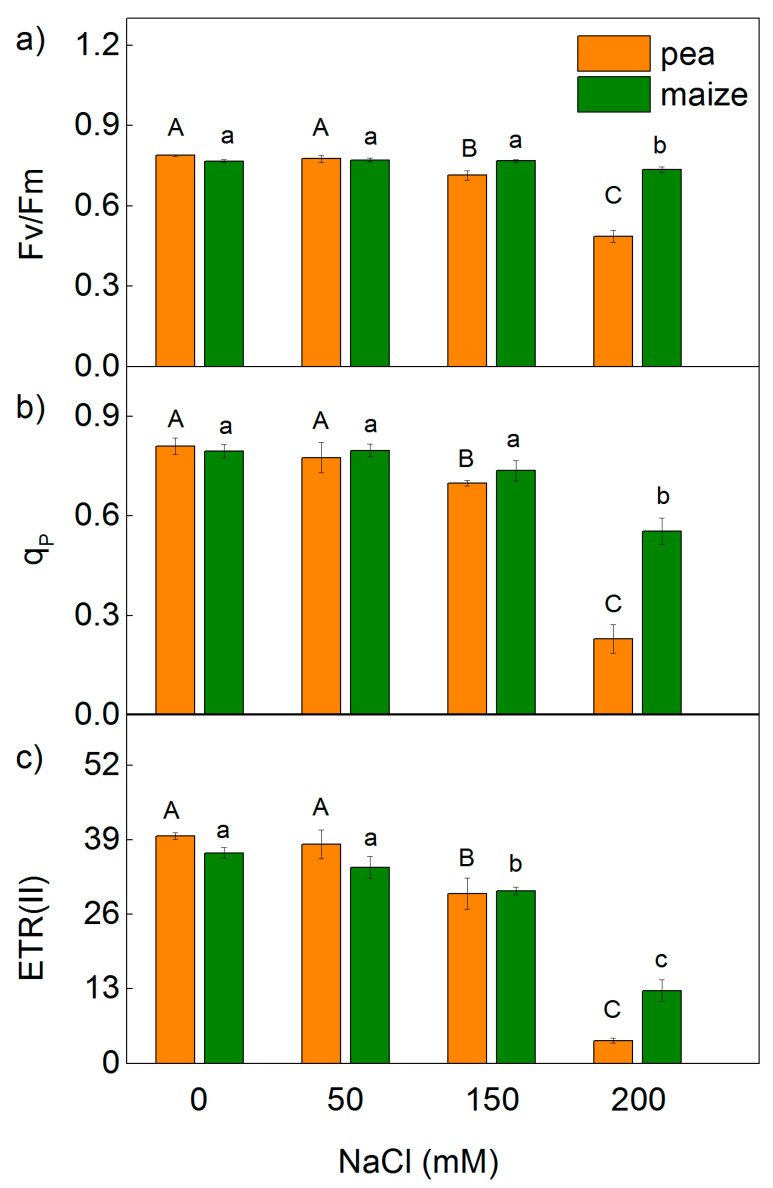
Effects of different NaCl concentrations on the selected parameters of PAM chlorophyll fluorescence in leaves of pea and maize: (**a**) the maximal quantum yield in dark-adapted state (Fv/Fm); (**b**) the photochemical quenching (q_p_); (**c**) PSII based electron transport rate (ETR(II)). The parameters are in relative units. Mean values (±SE) are from 10 independent measurements. Different letters indicate significant differences for the respective parameters at *p* < 0.05 (uppercase for pea and lowercase for maize).

**Figure 2 ijms-23-03768-f002:**
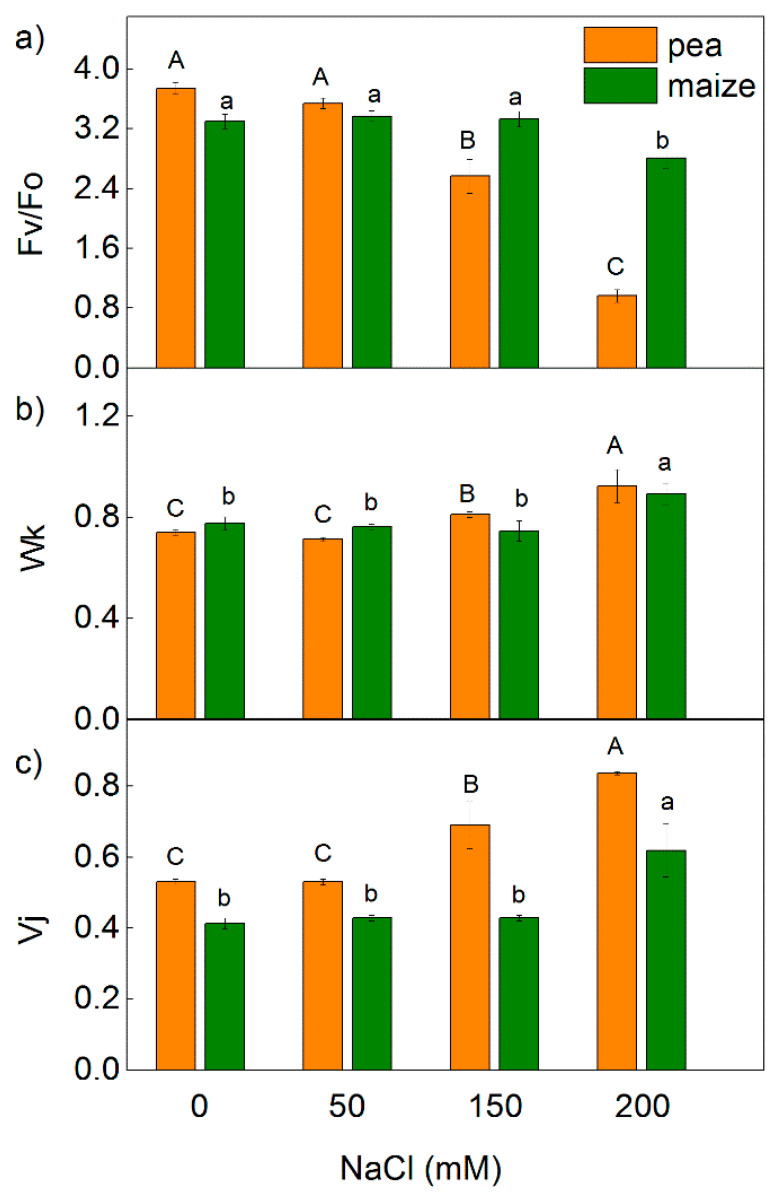
Effects of different NaCl concentrations on the selected parameters of chlorophyll fluorescence (Table 1) in leaves of pea and maize: (**a**) the ratio of photochemical to non-photochemical processes in PSII (Fv/Fo); (**b**) JIP parameter Wk; (**c**) JIP parameter Vj. The parameters are in relative units. Mean values (±SE) are from 10 independent measurements. Different letters indicate significant differences for the respective parameters at *p* < 0.05 (uppercase for pea and lowercase for maize).

**Figure 3 ijms-23-03768-f003:**
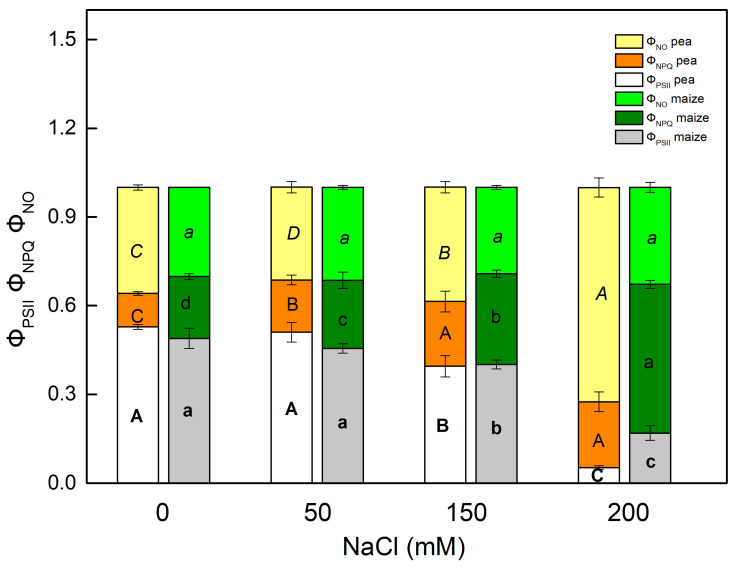
Effects of different NaCl concentrations in leaves of pea and maize on the effective quantum yield of a photochemical energy conversion of PSII (Φ_PSII_), the regulated (Φ_NPQ_) and non-regulated (Φ_NO_) energy loss in PSII. The parameters are in relative units. Mean values (±SE) are from 10 independent measurements. Different letters indicate significant differences for the respective parameters at *p* < 0.05 (uppercase for pea and lowercase for maize).

**Figure 4 ijms-23-03768-f004:**
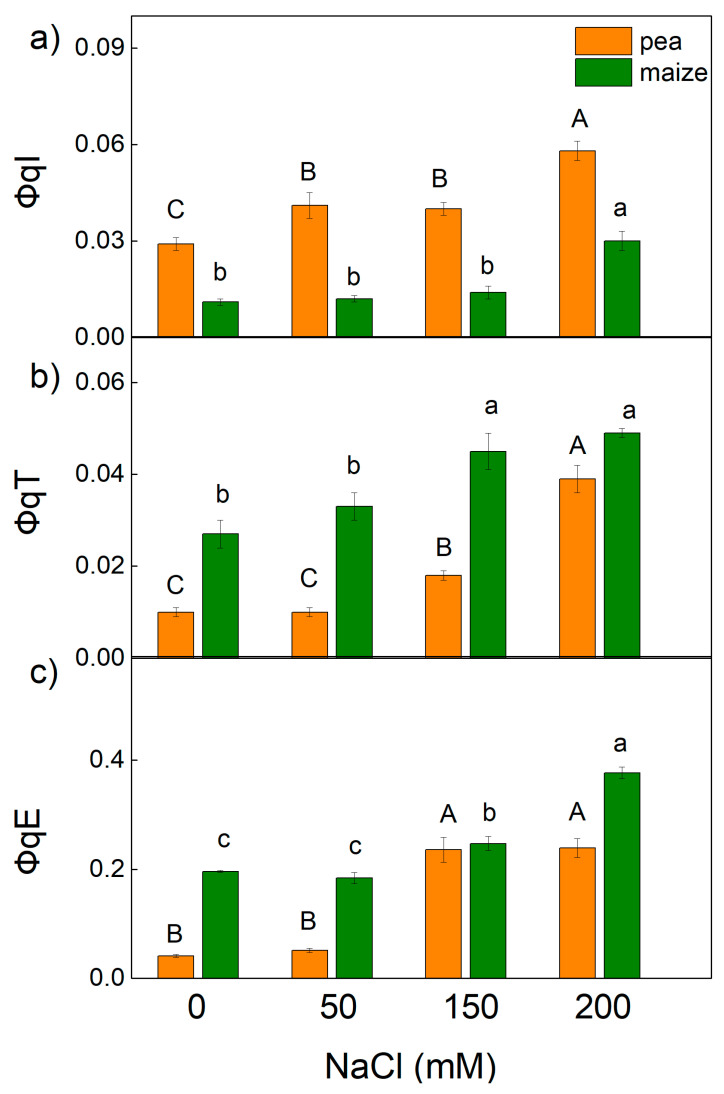
Effects of different NaCl concentrations on the quantum yields of NPQ components: (**a**) ΦqI (photoinhibitory component), (**b**) ΦqT (state transition component) and (**c**) ΦqE (energy-dependent component) in leaves of pea and maize. The parameters are in relative units. Mean values (±SE) are from 10 independent measurements. Different letters indicate significant differences for the respective parameters at *p* < 0.05 (uppercase for pea and lowercase for maize).

**Figure 5 ijms-23-03768-f005:**
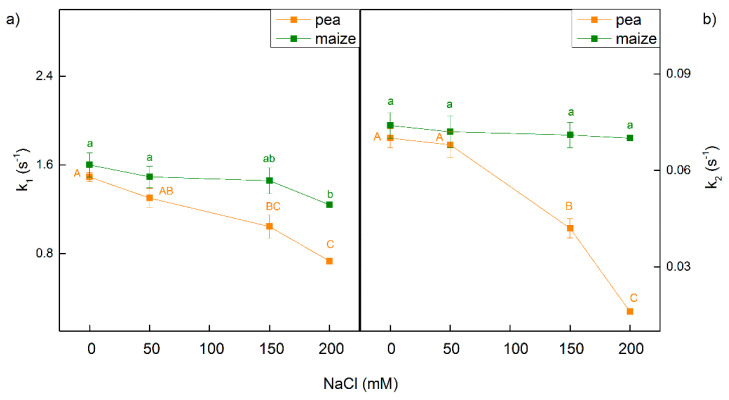
Effects of different NaCl concentrations on the dark relaxation of chlorophyll fluorescence induced by a single saturating light pulse in leaves of pea and maize. (**a**) Constant of the fast component (k_1_); (**b**) constant of the slow component (k_2_). Mean values (±SE) are from 10 independent measurements. Different letters indicate significant differences for the respective parameters at *p* < 0.05 (uppercase for pea and lowercase for maize).

**Figure 6 ijms-23-03768-f006:**
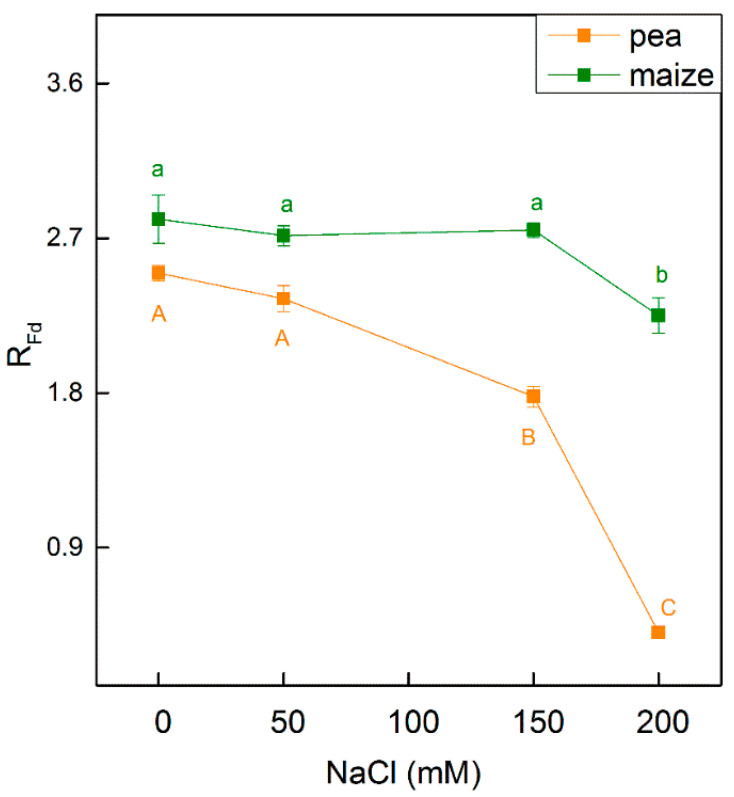
Effects of different NaCl concentrations on the chlorophyll fluorescence decay ratio R_Fd_ in leaves of pea and maize. The parameter is in relative units. Mean values (±SE) are from 10 independent measurements. Different letters indicate significant differences for the respective parameters at *p* < 0.05 (uppercase for pea and lower case are for maize).

**Figure 7 ijms-23-03768-f007:**
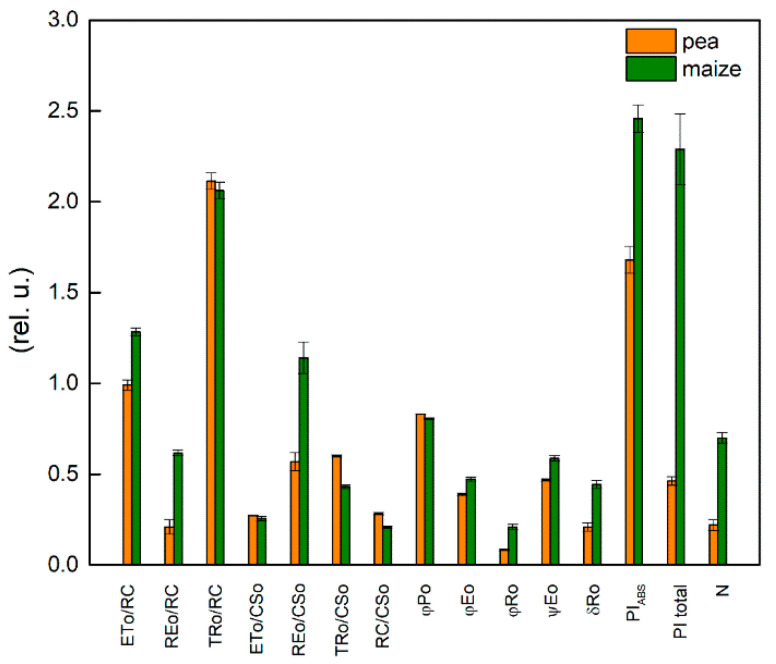
Selected JIP parameters (Table 1) were measured in leaves of pea and maize under physiological conditions. Mean values (±SE) were calculated from 20 independent measurements.

**Figure 8 ijms-23-03768-f008:**
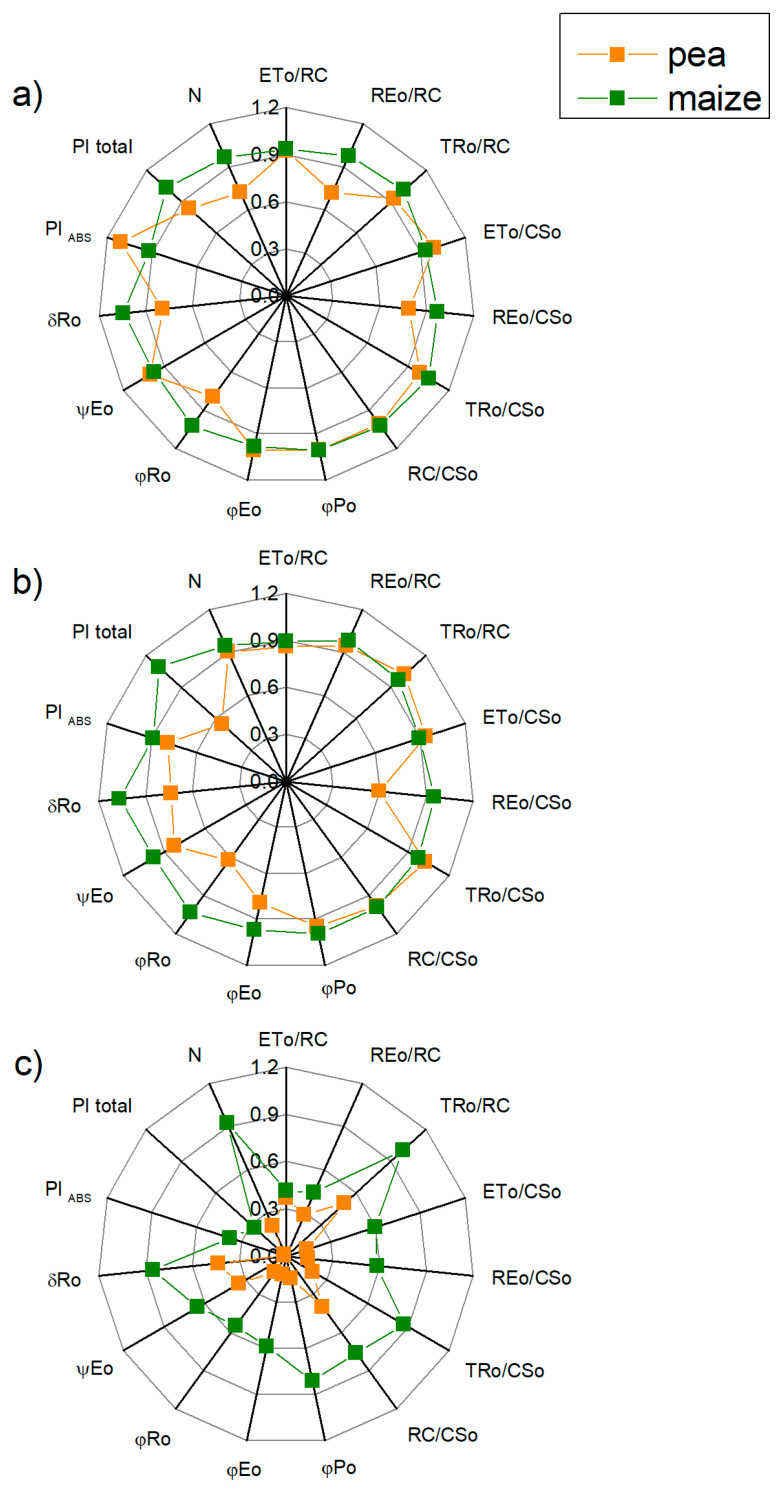
Effects of different NaCl concentrations on selected OJIP parameters (Table 1) in leaves of pea and maize plants grown in 50 mM NaCl (**a**), 150 mM NaCl (**b**) and 200 mM NaCl (**c**). The parameters are normalized to the respective control. Mean values (±SE) are from 20 independent measurements.

**Figure 9 ijms-23-03768-f009:**
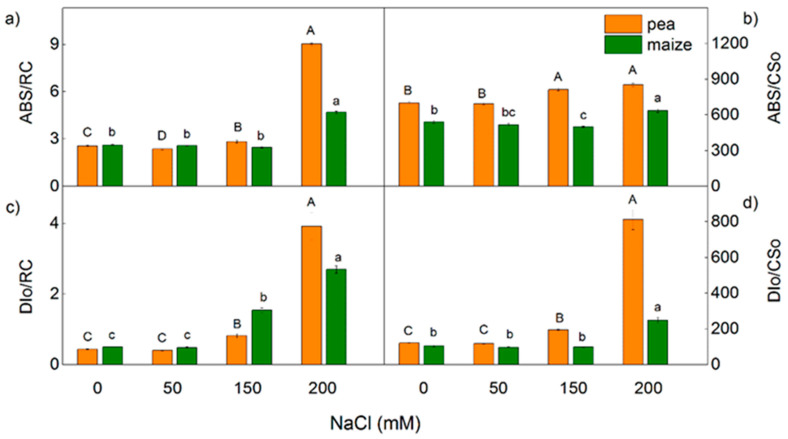
Effects of different NaCl concentrations on the selected JIP parameters (Table 1) in leaves of pea and maize. (**a**) absorption flux per RC (ABS/RC), (**b**) light energy absorption flux per CS (ABS/CSo), (**c**) dissipated energy flux per RC (DIo/RC), (**d**) dissipated energy flux per CS (DIo/CSo). The parameters are in relative units. Mean values (±SE) are from 20 independent measurements. Different letters indicate significant differences for the respective parameters at *p* < 0.05 (uppercase for pea and lowercase for maize).

**Figure 10 ijms-23-03768-f010:**
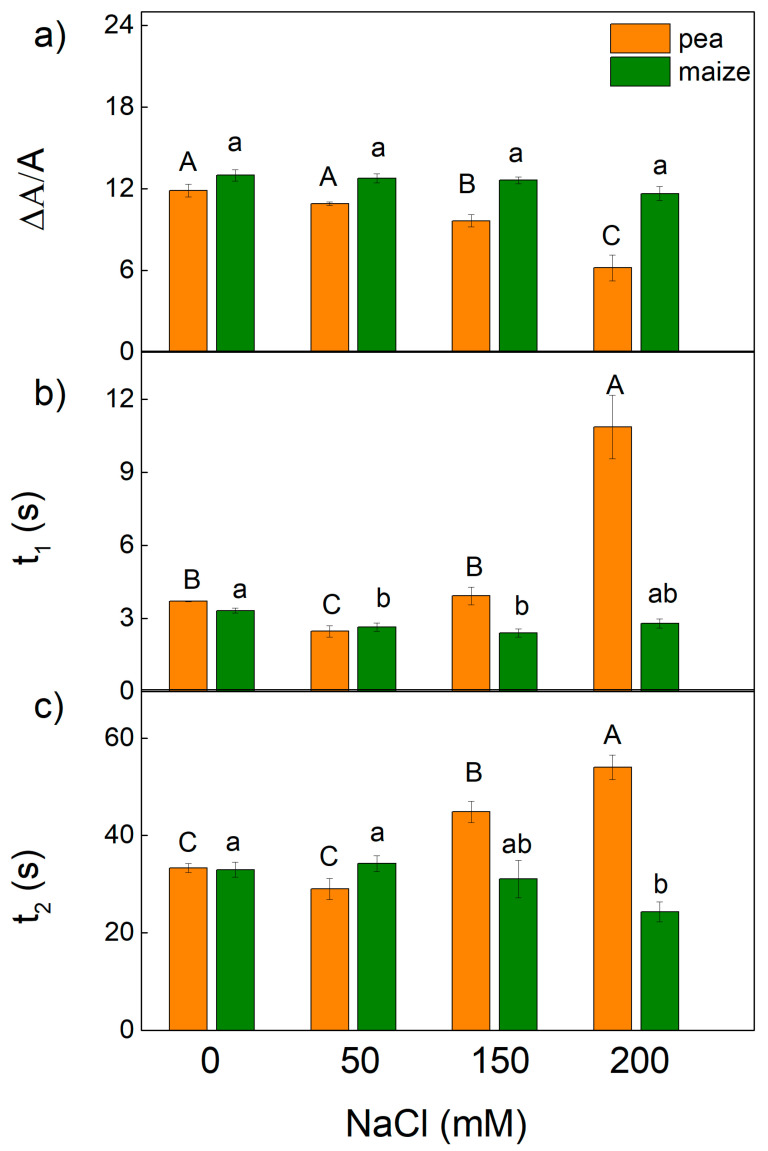
Effects of different NaCl concentrations on the far-red light-induced oxidation of P_700_ (ΔA/A) (the parameter is in relative units) (**a**), and the times of fast *t*_1_ (**b**) and slow *t*_2_ (**c**) of two components of dark reduction in the P_700_^+^ in leaves of pea and maize. Mean values (±SE) are from 10 independent measurements. Different letters indicate significant differences for the respective parameters at *p* < 0.05 (uppercase for pea and lowercase for maize).

**Table 1 ijms-23-03768-t001:** Description of the selected parameters of chlorophyll fluorescence, based on information presented in [40,41,42,43,44,45,46,47,48]. All parameters are in relative units.

JIP Parameters	
ABS/RC	Absorption flux per RC (apparent antenna size of an active RC)
ETo/RC	Electron transport flux (further than Q_A_^−^) per RC
REo/RC	Electron flux reducing end electron acceptors at the PSI acceptor side per RC
TRo/RC	Trapping flux (leading to Q_A_ reduction) per RC
DIo/RC	Dissipated energy flux per RC (at t = 0)
RC/ABS	The numbers of active RC per PSII antenna chlorophyll
ABS/CSo	Light energy (photons) absorption flux per cross section
ETo/CSo	Electron transport flux from Q_A_ to Q_B_ per cross section
REo/CSo	Electron transport flux until PSI acceptors per cross section
TRo/CSo	Maximum trapped exciton flux per cross section
DIo/CSo	Dissipated energy flux per cross section at t = 0
RC/CSo	Density of RCs (Q_A_ reducing PSII RC)
φPo	Maximum quantum yield of primary photochemistry (at t = 0)
φEo	Quantum yield of electron transport (at t = 0)
φRo	Quantum yield of reduction in end electron acceptors at the PSI acceptor side
ψEo	Moves an electron into the electron transport chain beyond Q_A_^−^
δRo	Efficiency/probability with which an electron from the intersystem electron carriers moves to reduce end electron acceptors at the PSI acceptor side
PI_ABS_	Performance index (potential) for energy conservation from exciton to the reduction in intersystem electron acceptors
PI total	Performance index (potential) for energy conservation from exciton to the reduction in PSI end acceptors
N	Maximum turnovers of Q_A_ reduction until Fm was reached
Vj	Relative variable fluorescence at the J step
Wk	The ratio of the K phase to the J phase
PAM parameters	
Fv/Fm	The maximum quantum yields of primary photochemistry of PSII
Fv/Fo	The ratio of photochemical to non-photochemical processes in PSII
q_p_	The photochemical quenching
ETR(II)	PSII based electron transport rate
Φ_PSII_	The photochemical energy conversion in PSII
Φ_NPQ_	The quantum yields of regulated energy losses in PSII
Φ_NO_	The quantum yields of non-regulated energy losses in PSII
R_Fd_	The fluorescence decrease from Fm to a steady state chlorophyll fluorescence after continuous saturated illumination

**Table 2 ijms-23-03768-t002:** Effects of different NaCl concentrations on the components of the performance indices PI_ABS_ and PI total in leaves of pea and maize. The parameters are in relative units. Mean values (±SE) are from 20 independent measurements. Different letters indicate significant differences for the respective parameters at *p* < 0.05 (uppercase for pea and lowercase for maize).

	RC/ABS	φPo/(1 − φPo)	ψEo/(1 − ψEo)	δREo/(1 − δREo)
pea				
control	0.393 ± 0.008 ^A^	4.837 ± 0.077 ^A^	0.883 ± 0.025 ^A^	0.272 ± 0.054 ^A^
50 mM NaCl	0.425 ± 0.010 ^A^	4.908 ± 0.084 ^A^	0.889 ± 0.028 ^A^	0.201 ± 0.030 ^A^
150 mM NaCl	0.356 ± 0.012 ^B^	3.793 ± 0.034 ^B^	0.668 ± 0.109 ^B^	0.245 ± 0.056 ^A^
200 mM NaCl	0.112 ± 0.004 ^C^	1.360 ± 0.005 ^C^	0.484 ± 0.025 ^C^	0.100 ± 0.010 ^B^
maize				
control	0.392 ± 0.010 ^a^	4.174 ± 0.101 ^a^	1.440 ± 0.090 ^a^	0.801 ± 0.071 ^a^
50 mM NaCl	0.393 ± 0.013 ^a^	4.300 ± 0.066 ^a^	1.341 ± 0.042 ^a^	0.880 ± 0.086 ^a^
150 mM NaCl	0.403 ± 0.004 ^a^	4.079 ± 0.119 ^a^	1.341 ± 0.040 ^a^	0.925 ± 0.103 ^a^
200 mM NaCl	0.313 ± 0.043 ^b^	2.582 ± 0.009 ^b^	0.730 ± 0.095 ^b^	0.619 ± 0.067 ^b^

## Data Availability

Not applicable.

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
