# Peer review of "Assessment of the Photosynthetic Apparatus Functions by Chlorophyll Fluorescence and P700 Absorbance in C3 and C4 Plants under Physiological Conditions and under Salt Stress"

_ijms, 2022, doi:10.3390/ijms23073768_

Round 1

Reviewer 1 Report

The manuscript entitled “Monitoring of the Photosynthetic Apparatus Functions by Chlorophyll Fluorescence and P700 Absorbance in C3 and C4 3 Plants at Physiological Conditions and under Salt Stress" is based on original research experiment and the presented results therein broaden the knowledge in the field of applied plant science and plant physiology. Authors aimed to evaluated the effects of different concentrations of NaCl on the primary processes of photosynthesis in pea (Pisum sativum L.) and in maize (Zea mays L.). The scope of work includes the performance of experiment in controlled conditions, during which measurements of pulse-amplitude-modulated (PAM) chlorophyll a fluorescence, prompt fluorescence (JIP test) and P700 photooxidation were obtained.

There is no doubt that this work is in the scope of International Journal of Molecular Sciences. The publication presents generally interesting and important studies. The paper is well organized, presented in a logical sequence, and has adequate bibliographic review. The work delivers interesting results and can be the important source of valuable information.

The introduction is properly composed. The materials and methods section contains the basic requested elements and provide information about the experimental preparations, analyses and growth conditions. The data analysis is properly provided. The results show valuable information. The obtained data are discussed sufficiently. The strength of the work is the combination of two techniques for measuring chlorophyll fluorescence. In majority of works, the emphasis is usually on just one of them. However, their combination makes it possible to obtain a broader view of the influence of the stress factor on the photosynthetic apparatus

However, authors made few shortcomings that must be corrected before the publication of the work:

  • Title: I am not convinced that the word "monitoring" should be used in the title of the research paper. The title should outline the research problem, and monitoring is not.
  • The “OJIP” abbreviation is used for induction curve. When we wrote about the method or parameters, “JIP-test” phrase should be used.
  • ChFl parameters are in relative units, and it should be marked on OY axis on figures.

I would like to underline that my remarks are auxiliary and not undertake the quality and importance of the paper.

Author Response

Report to the comments of reviewer 1 on manuscript titled  “Monitoring of the photosynthetic apparatus functions by chlorophyll fluorescence and P700 absorbance in C3 and C4 3 plants at physiological conditions and under salt stress" ” by Stefanov et al.

The authors would like to thank the reviewer for constructive and insightful comments in relation to this work. We considered all comments and suggestions to be justified, and corrected the manuscript accordingly. Please, find the detailed list of all edits below. The newly edited text parts are indicated with red letter.

Comments and Suggestions for Authors

The manuscript entitled “Monitoring of the Photosynthetic Apparatus Functions by Chlorophyll Fluorescence and P700 Absorbance in C3 and C4 3 Plants at Physiological Conditions and under Salt Stress" is based on original research experiment and the presented results therein broaden the knowledge in the field of applied plant science and plant physiology. Authors aimed to evaluated the effects of different concentrations of NaCl on the primary processes of photosynthesis in pea (Pisum sativum L.) and in maize (Zea mays L.). The scope of work includes the performance of experiment in controlled conditions, during which measurements of pulse-amplitude-modulated (PAM) chlorophyll a fluorescence, prompt fluorescence (JIP test) and P700 photooxidation were obtained.

There is no doubt that this work is in the scope of International Journal of Molecular Sciences. The publication presents generally interesting and important studies. The paper is well organized, presented in a logical sequence, and has adequate bibliographic review. The work delivers interesting results and can be the important source of valuable information.

The introduction is properly composed. The materials and methods section contains the basic requested elements and provide information about the experimental preparations, analyses and growth conditions. The data analysis is properly provided. The results show valuable information. The obtained data are discussed sufficiently. The strength of the work is the combination of two techniques for measuring chlorophyll fluorescence. In majority of works, the emphasis is usually on just one of them. However, their combination makes it possible to obtain a broader view of the influence of the stress factor on the photosynthetic apparatus

However, authors made few shortcomings that must be corrected before the publication of the work:

Title: I am not convinced that the word "monitoring" should be used in the title of the research paper. The title should outline the research problem, and monitoring is not.

Answer: The title has been changed in the revised manuscript.

The “OJIP” abbreviation is used for induction curve. When we wrote about the method or parameters, “JIP-test” phrase should be used.

Answer: Thank you for your helpful comments. The corrections have been made in the revised manuscript.

ChFl parameters are in relative units, and it should be marked on OY axis on figures.

Answer: We agree that the parameters of Chl fluorescence are in relative units. In the revised manuscript we wrote that the parameters are in relative units in the legend of Table 1, as well as under the figures. If you think that it is better to have the inscription (relative units) on the axis, we will add relative units to the axis.

Sincerely yours,

Dr. Emilia Apostolova

Reviewer 2 Report

The manuscript entitled “Monitoring of the photosynthetic apparatus….” comprehensively presents the activity of the photosynthetic apparatus in selected C3 (Pisum sativum) and C4 (Zea mays) plant species, mainly based on chlorophyll fluorescence measurements with PSII. In addition, the activity of PSII and PSI in response to salinity is compared. The results indicate that the photosynthetic apparatus in P. sativum is more sensitive to salt (NaCl) than in Z. mays. Therefore, the presented work is interesting and could be of interest to a wide range of IJMS readers.

Nevertheless, some errors and inaccuracies were found in the manuscript that need to be clarified and/or corrected.

All detailed comments are listed below.

The text of the manuscript should be checked by a native speaker for linguistic accuracy.

p. 1, l. 39: Enter the abbreviation "ROS" and use it consistently throughout the manuscript.

p. 4, l. 52 -58: This fragment contains textbook information on C3 and C4 photosynthesis. In my opinion, it should be removed or shortened.

p. 2, l. 97 and in many other places: To my knowledge, PAM estimates ETR near PSII, i.e. ETR(II) or PSI, i.e. ETR(I). The information in "Materials and Methods" (p. 15) indicates that it is ETR(II). This should be clarified.

p. 3, l. 102, 104: Fv/Fm, qp, Fv/Fo. I think these fluorescent parameters should also be briefly described in Table 1.

p. 8, Fig. 6. The description of Fig. 6 should indicate that these are measurements under control conditions (without NaCl).

p. 13, l. 341-344 and 353 - 355: It is not clear to me why the authors discuss fluorescence parameters such as Fv/Fm in relation to Chla and Chlb, since the content of these pigments was not measured at all in the experiments presented in this manuscript (?). This should be changed.

p. 13, l. 374 – 379: If Wk and Fv/Fo indicate OEC activity, reconsider presenting these results in a single figure.

p. 14, l. 403 - 404: Remove the sentence: “Salt treatment...”

p. 14, l. 405 - 409: Some citations concerning interpretation of the fluorescent parameters: qE, qT, qI, NPQ should be added.

In general, the "Discussion" contains some repetition of the description of the results, which is partly a repetition of what is contained in the "Results" section.

p. 15, l. 490: “Conclusions” after "Materials and Methods. Is this OK?

p. 15 and 16: In my opinion, the first 3-4 sentences do not fit the "Conclusions" and should be thoroughly rewritten or removed.

p. 16, l. 505: “indices” not “induces”

Author Response

Report to the comments of reviewer 2 on manuscript titled  “Monitoring of the photosynthetic apparatus functions by chlorophyll fluorescence and P700 absorbance in C3 and C4 3 plants at physiological conditions and under salt stress" ” by Stefanov et al.

The authors would like to thank the reviewer for constructive and insightful comments in relation to this work. We considered all comments and suggestions to be justified, and corrected the manuscript accordingly. Please, find the detailed list of all edits below. The newly edited text parts are indicated with red letter.

The manuscript entitled “Monitoring of the photosynthetic apparatus….” comprehensively presents the activity of the photosynthetic apparatus in selected C3 (Pisum sativum) and C4 (Zea mays) plant species, mainly based on chlorophyll fluorescence measurements with PSII. In addition, the activity of PSII and PSI in response to salinity is compared. The results indicate that the photosynthetic apparatus in P. sativum is more sensitive to salt (NaCl) than in Z. mays. Therefore, the presented work is interesting and could be of interest to a wide range of IJMS readers.

Nevertheless, some errors and inaccuracies were found in the manuscript that need to be clarified and/or corrected.

All detailed comments are listed below.

The text of the manuscript should be checked by a native speaker for linguistic accuracy.

Answer: The manuscript was edited for the English language. Due to these extensive edits, the track-changes were not applied. Instead, the newly edited text parts with more changes are indicated with red letters.

  1. 1, l. 39: Enter the abbreviation "ROS" and use it consistently throughout the manuscript.

Answer: The corrections have been made in the revised manuscript.

  1. 4, l. 52 -58: This fragment contains textbook information on C3 and C4 photosynthesis. In my opinion, it should be removed or shortened.

Answer: The corrections have been made in the revised manuscript.

  1. 2, l. 97 and in many other places: To my knowledge, PAM estimates ETR near PSII, i.e. ETR(II) or PSI, i.e. ETR(I). The information in "Materials and Methods" (p. 15) indicates that it is ETR(II). This should be clarified.

Answer: The proposed clarifications are made in the text and in the figures.

  1. 3, l. 102, 104: Fv/Fm, qp, Fv/Fo. I think these fluorescent parameters should also be briefly described in Table 1.

Answer: The PAM parameters are briefly described in Table 1 in the revised manuscript

  1. 8, Fig. 6. The description of Fig. 6 should indicate that these are measurements under control conditions (without NaCl).

Answer: The corrections have been made in the revised manuscript.

  1. 13, l. 341-344 and 353 - 355: It is not clear to me why the authors discuss fluorescence parameters such as Fv/Fm in relation to Chla and Chlb, since the content of these pigments was not measured at all in the experiments presented in this manuscript (?). This should be changed.

Answer: The effects of the different NaCl concentrations on the amount of chlorophyll and the chl a/b ratio is given in the Table in the supplementary material (Table S1).

  1. 13, l. 374 – 379: If Wk and Fv/Fo indicate OEC activity, reconsider presenting these results in a single figure.

Answer: The parameters Wk and Fv/Fo are given in one figure in the revised manuscript (Figure 2).

  1. 14, l. 403 - 404: Remove the sentence: “Salt treatment...”

Answer: The correction has been made in the revised manuscript.

  1. 14, l. 405 - 409: Some citations concerning interpretation of the fluorescent parameters: qE, qT, qI, NPQ should be added.

Answer: New references have been added to the revised manuscript (77, 78, 79).

In general, the "Discussion" contains some repetition of the description of the results, which is partly a repetition of what is contained in the "Results" section.

Answer: The discussion was rewritten.  In the revised manuscript  the passages with more significant corrections they are marked in red letters.

  1. 15, l. 490: “Conclusions” after "Materials and Methods. Is this OK?

Answer: According to the template of the journal, the Conclusion is after Materials and Methods.

  1. 15 and 16: In my opinion, the first 3-4 sentences do not fit the "Conclusions" and should be thoroughly rewritten or removed.

Answer: Corrections have been made to the revised manuscript.

  1. 16, l. 505: “indices” not “induces”

Answer: Corrections have been made to the revised manuscript.

Sincerely yours,

Dr. Emilia Apostolova